# Biasing Deep ConvNets for Semantic Segmentation of Medical Images with a Prior-driven Prediction Function

**Olivier Petit**[1,2]                                    olivier.petit@visiblepatient.com

**Nicolas Thome**[2]                                         nicolas.thome@cnam.fr

**Luc Soler**[3]                                               luc.soler@ircad.fr

[1] *Visible Patient, Strasbourg*

[2] *CNAM, Paris*

[3] *IRCAD, Strasbourg*

## 1. Introduction

Organ segmentation in CT-scans is a major issue in medical image analysis. Recently deep learning methods have brought impressive results for the task of semantic segmentation (Chen et al., 2018; Ronneberger et al., 2015; Long et al., 2015). However it remains very challenging especially due to the visual ambiguities (Kohl et al., 2018). The local visual context is insufficient and including external knowledge could help obtaining a better segmentation.

In this paper we address the problem of including prior information about the shape and spatial position of the organs to improve the performance of semantic segmentation. It is particularly relevant in medical imaging where there is conventions on the structure of the images. For example, the position of the patient is always the same (spine at the bottom). The main changes reside on the patient anatomy which slightly varies from one to another.

Including such a knowledge by biasing the predictions of Fully Convolutional Networks (FCN) is a difficult problem. Due to weight sharing and pooling operations, FCN models focus on local structures and loose the input position information. Moreover, it has also been highlighted that ConvNets have difficulties succeeding at simple spatial prediction tasks like coordinate transform (Liu et al., 2018).

In the medical imaging literature, some methods have been proposed to incorporate such kind of information. For example cascaded networks (Roth et al., 2018) rely on selecting a Region of Interest (RoI) by a first model, which is then finely segmented by a second model. Those methods are efficient but rely on the good localization of the RoI. As a consequence, each step limits the overall performances.

Other methods try to incorporate spatial information implicitly. For instance by generating distance maps with a GAN (Trullo et al., 2019) or by learning attention maps through an attention mechanism (Oktay et al., 2018). In those approaches, the spatial information is learned during training so we have no control of what is actually learned and how it biased the prediction.

Our proposed approach incorporates the prior knowledge more explicitly in a late fusion operation. We first extract the prior knowledge from the training set and then add it to

the model thanks to a prior-driven softmax function which includes the prior directly into the softmax at the end of the network. By doing so, we keep full control on how the prior is built and how it participates on the learning.

## 2. Learning Deep ConvNets with Spatial Priors

The main idea of our method is to take advantage of the prior knowledge on the position and shape of the organ we want to segment in the image. As shown in figure 1, the input image goes through a FCN (It could be any architecture) which outputs a prediction for every pixels. The result is merged with a prior map through a custom operation we called *prior-driven softmax* function. At the end, the final result is a segmentation map biased by the prior map.

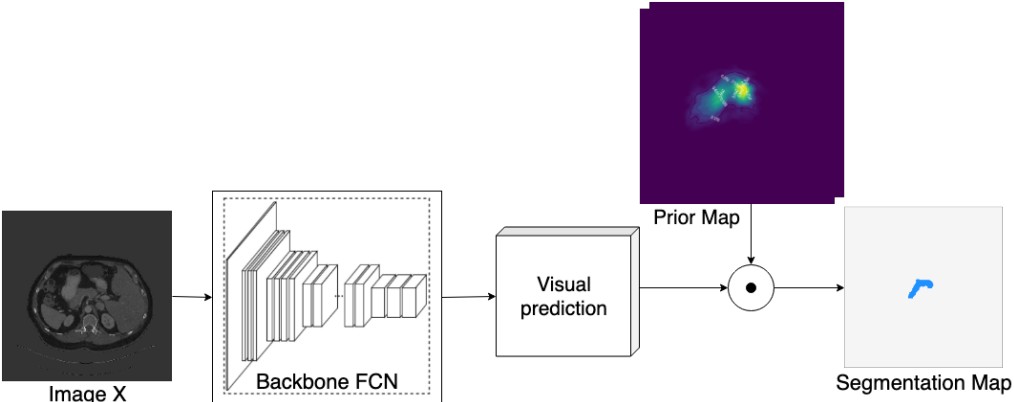

Figure 1: Proposed approach

**Spatial prior**   First we model the prior information as a probability map. It is computed on the training annotations by a histogram estimation. We first assume that all images are roughly registered in the (x,y) plane for a given z coordinate. Then the spatial prior map is computed by accumulating the annotations for each class and then by normalizing the histogram such that $\sum_{k=1}^{K} p_k = 1$.

An example of a prior map for the pancreas is shown in figure 1. We can see how the prior is localized in a very specific region of the image, here for the pancreas.

**Prior-driven softmax prediction**   In order to biased the prediction of the FCN, we incorporate the prior into the model with the prior-driven softmax. It can be seen as an extension of the classic softmax activation function where we multiply the exponentials by our prior as shown in equation 1.

$$s_k = \frac{e^{\tilde{s}_k} \; p_k}{\sum_{c=1}^{K} e^{\tilde{s}_c} \; p_c} \tag{1}$$

One can notice that when the prior is uniformly distributed, i.e. $p_k = p_c = \frac{1}{K} \; \forall k \in \{1..K\}$, the prior-driven softmax reduces to the standard softmax function. On the other

hand, when the prior is not uniform, it can be used to bias the prediction of a given class $k$ depending on its spatial location. For example, if $p_k$ is close to 1 (resp. 0), the prediction of class $k$ is made close to 1 (resp. 0) whatever the $e^{\tilde{s}_k}$ value. Our prior-driven softmax prediction function in equation 1 can thus be leveraged to overcome visual ambiguities between organs and the background.

## 3. Experiments

We perform experiments on the publicly available TCIA pancreas dataset which is composed of 82 CT-scans where the pancreas has been manually segmented (Holger R. Roth and Summers, 2016). 80% of the data are used for training and 20% for testing. We evaluate the performances using the three metrics shown in Table 1. We can observe a consistent gain of the proposed method over the FCN baseline based on a ResNet-101 (He et al., 2016), *e.g.* 2.03 pt improvement on global Dice. This result is very encouraging, especially with the simple 2D prior used in these experiments.

Table 1: Results obtained on the Pancreas TCIA dataset.

| Model | Dice per case | Dice global | ASSD (mm) |
|---|---|---|---|
| FCN (ResNet-101) | 0.7013 | 0.7021 | 2.7375 |
| FCN (ResNet-101) + Prior | **0.7216** | **0.7251** | **2.4539** |

Qualitatively, we observe that our method enables us to ignore irrelevant regions of the image and thus focus on the most probable ones. Figure 2 is an example where our method successfully leverages the prior to recover the pancreas region that the baseline misses.

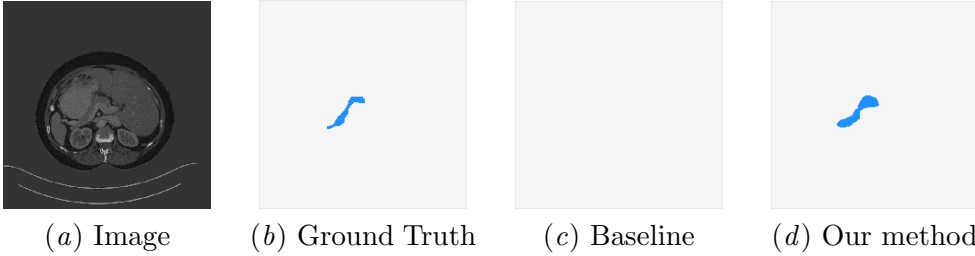

$(a)$ Image        $(b)$ Ground Truth        $(c)$ Baseline        $(d)$ Our method

Figure 2: Qualitative results

## 4. Conclusion and perspectives

In this paper we proposed to use prior knowledge on the shape and absolute position of the organ to segment it. Next we plan to try other architectures like U-Net (Ronneberger et al., 2015) but also other datasets with multiple organs. Then, the 2D prior map could be extended to a 3D map which will be obviously more accurate through the slices, and thus brought more precise information. Finally it could be interesting to adapt models which exploit contextual information like in (Durand et al., 2015).

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
