# OpenReview forum: "Biasing Deep ConvNets for Semantic Segmentation of Medical Images with a Prior-driven Prediction Function"
_MIDL.io/2019/Conference/Abstract — MIDL Abstract 2019_

### Official Review · AnonReviewer2 · 2019-04-30
**Simple method of including anatomical priors into anatomical segmentation problems**

**Rating:** 3
**Confidence:** 3

**Review:**

This abstract describes a simple method of including prior anatomical knowledge into semantic segmentation problems by including an average shape into the final softmax of a convoluational neural network.

The method is only evaluated against a version without using this prior, but not against other methods of including a prior. For future work, if the authors are not aware of those works already it is suggested to have a look at the following. Oktay et al. "Anatomically constrained neural networks (ACNNs): application to cardiac image enhancement and segmentation" and Dalca et al. "Anatomical Priors in Convolutional Networks for Unsupervised Biomedical Segmentation" (and other recent works by Dalca et al.).

Despite not comparing against related work, this simple method seems to lead to quite substantial improvements on the investigated task and offers an interesting proof-of-concept.

How much of a limitation is the fact that the technique requires images to be roughly registered?

---

### Official Review · AnonReviewer1 · 2019-05-02

**Rating:** 3
**Confidence:** 3

**Review:**

The paper proposed a method to incorporate the prior information into deep neural network. The way to do this is to add the prior to the softmax activation function. I see both innovation in the methods and solid validation results.

I would suggest the authors discussion the mpact of the quality of the registration how robust this architecture is.

---

### Decision · Program_Chairs · 2019-05-06
**Acceptance Decision**

Accept